# Effects of Autolyzed Yeast Supplementation in a High-Starch Diet on Rumen Health, Apparent Digestibility, and Production Variables of Lactating Holstein Cows

**DOI:** 10.3390/ani12182445

**Published:** 2022-09-16

**Authors:** Sara E. Knollinger, Milaine Poczynek, Bryan Miller, Isabel Mueller, Rodrigo de Almeida, Michael R. Murphy, Felipe C. Cardoso

**Affiliations:** 1Department of Animal Sciences, University of Illinois, Urbana, IL 61801, USA; 2Department of Animal Sciences, Universidade Federal do Paraná, Curitiba 80035-050, PR, Brazil; 3BIOMIN America Inc., Overland Park, KS 66210, USA; 4BIOMIN Holding GmbH, 3131 Getzersdorf, Austria

**Keywords:** apparent digestibility, feed efficiency, N excretion, rumen pH, starch, yeast

## Abstract

**Simple Summary:**

Feeding high-starch (HS) diets has negative implications on the ruminal health and milk production of dairy cattle. Autolyzed yeast (AY) supplementation may alleviate the negative effects of an HS diet fed to dairy cows. Therefore, the objectives of this study were to determine the effects of the supplementation of commercially available AY in HS diets on lactation performance, rumen pH, N utilization, apparent digestibility (AD), and blood metabolites in lactating Holstein cows. This study postulated that AY supplementation may improve the rumen environment (rumen pH and AD) when feeding HS diets, thus improving lactation performance and altering blood metabolites.

**Abstract:**

Fifteen multiparous rumen-cannulated Holstein cows were assigned to one of five treatments in a replicated 5 × 5 Latin square design. The treatments were low-starch (**LS**) (22.8 ± 1% of dry matter; **DM**) without autolyzed yeast (**AY**; **LS0**, control), high-starch (HS) (31.2 ± 4% of DM) without AY (**HS0**), and HS with either 15 g (**HS15**), 30 g (**HS30**), or 45 g (**HS45**) of AY supplementation. Cows in HS0 had increased (*p* < 0.03) dry matter intake (**DMI**; 24.9 kg/d) and energy-corrected milk (**ECM**; 34.4 kg/d) compared to cows in LS0 (19.9 and 31.3 kg/d, respectively). There was a tendency for a quadratic treatment effect for feed efficiency (ECM/DMI, *p* = 0.07) and crude protein (**CP**) apparent digestibility (**AD**) (*p* = 0.09). Cows in HS45 tended (*p* = 0.09) to have increased DMI (25.6 kg/d) compared to cows in HS0 (24.9 kg/d). Cows in HS0 had greater (*p* < 0.04) milk protein nitrogen (**N**; 166 g/d) and microbial N production (161 g/d) than those in LS0 (140 and 138 g/d, respectively). In conclusion, the addition of AY tended to improve DMI, feed efficiency, and CP AD when cows were fed the HS diet.

## 1. Introduction

Increasing dietary starch concentration by feeding readily fermentable grains can mitigate the negative energy balance postpartum and improve cow performance. It is documented that feeding high-starch (**HS**) diets can result in greater fecal nitrogen (**N**) excretion compared to feeding low-starch (**LS**) diets [1]. Additionally, increased environmental regulations have created new pressures for nutrient use improvements in the agricultural industry [2]. The release of N into the environment can negatively affect human health through the combination of ammonia and other chemicals in the atmosphere [2]. Therefore, the excretion of nitrogen is becoming more of a concern on commercial dairy farms.

Typical HS diets for lactating dairy cows are between 26 and 32% starch in diet dry matter (**DM**; [3,4,5]). Dairy cattle often experience deleterious effects when fed highly fermentable diets for an extended period, exposing cows to a greater risk of ruminal acidosis [6]. Diets high in fermentable concentrates can alter rumen function, leading to increased acid production and reduced rumination and salivation, and result in decreased rumen pH and potential inflammation of the rumen epithelia [7,8]. Reduced dietary starch increases ruminal pH and acetate concentration when compared to HS diets [9]. However, feeding reduced starch diets may result in decreased milk yield.

Fecal and urinary nitrogen, through bacterial degradation, can be converted to ammonia. The environmental protection agency classifies ammonia as an air quality threat because it can contribute to nitrate contamination of groundwater and surface water eutrophication, and hinder air quality [10]. Dietary factors in the diet such as CP concentration, in addition to carbohydrate and forage type, can impact the quantity and form of N excretion, whether fecal or urinary [1].

A mitigation practice to help offset the negative effects of HS diets is the supplementation of yeast. Live yeast and yeast-based products supplemented in the diet of cattle, swine, lambs, and poultry have been reported to have numerous beneficial effects, in particular the *Saccharomyces cerevisiae* strain [11]. Different types of yeast products are defined based on their active ingredients and mode of action. Active live yeast is classified as fermentable, dried, and containing at least 15 × 10^9^ live yeast cells per gram [12]. Yeast is commonly included in diets for high-producing dairy cows because of its positive effects on milk production. In addition to the high availability of prebiotic bioactive compounds, including β-glucans and nutrients in autolyzed yeast (**AY**), other components of the yeast cell wall have been found to activate immune cells [13,14]. It has been well-documented that the inclusion of AY (*Saccharomyces cerevisiae*) in ruminant diets improves dry matter intake (**DMI**), rumen pH, volatile fatty acid (**VFA**) profile, and nutrient digestibility [7,15,16]. 

Supplementation of AY in highly fermentable diets may reduce ruminal lactate concentration, increasing ruminal pH and promoting lactation performance [17]. Additionally, AY may increase the fiber-adhering cellulolytic bacteria that promote fiber digestion and the growth of rumen bacteria [15,18]. Direct-fed microbials, such as yeast, can help stabilize rumen fermentation by increasing the beneficial bacteria, ruminal pH, and oxygen removal [19,20]. According to Julien et al. [21], yeast supplementation, in particular the *S. cerevisiae* strain, may reduce N excreted in the feces and CH_4_ production attributable to microbiome changes in the rumen [7,22,23]. Together, these findings indicate that AY supplementation of dairy cow diets could effectively enhance cellulolytic activity and thereby improve the rumen environment. Yet, the mode of action, effects of diet composition, and recommended concentration in the diet are not well-understood and vary among yeast products. 

Therefore, the objectives of this study were to determine the effects of the supplementation of commercially available AY in HS diets on lactation performance, rumen pH, N utilization, apparent digestibility (**AD**), and blood metabolites in lactating Holstein cows. This study postulated that AY supplementation may improve the rumen environment (rumen pH and AD) when feeding HS diets, thus improving lactation performance and altering blood metabolites.

## 2. Materials and Methods

### 2.1. Animal Care and Housing

All experimental procedures were approved by the University of Illinois (Urbana-Champaign) Institutional Animal Care and Use Committee (#17172). The experimental period was from January through April 2018. Cows were housed in tie stalls with sand bedding, fed ad libitum, and had free water access. Diets (total-mixed ration; **TMR**) were formulated using AMTS.Cattle.Pro version 4.7 (2017, AMTS, LLC, Groton, NY, USA) (Table 1) for cows at 70 days in milk (**DIM**), with 703 kg of body weight (**BW**), producing 41 kg of milk/d with a target of 3.8% milk fat and 3.2% milk protein, and a predicted DMI of 25 kg/d.

### 2.2. Experimental Design 

A total of fifteen healthy multiparous rumen-cannulated Holstein cows at the start of the experiment [BW (mean ± SD) = 623 ± 73 kg; DIM = 77 ± 26] were assigned to one of five treatments in a replicated 5 × 5 Latin square design. Periods (21 d) were divided into an adaptation phase (d 1 to 14) and a measurement phase (d 15 to 21). On d 1 to 21 of each period cows received one of five dietary treatments: LS diet (22.8 ± 0.7% starch of DM) without AY (**LS0**) or HS diet (31.2 ± 4.2% starch of DM) without AY (**HS0**), 15 g of AY (**HS15**), 30 g of AY (**HS30**), or 45 g of AY (**HS45**). The AY (*S. cerevisiae*) product was spray-dried and produced by an internal process technology for standardized autolytic degradation of the yeast cell (Levabon, BIOMIN Holding GmbH, Austria). As described by the manufacturer, the product’s chemical analysis is (DM = 96%; crude protein (**CP**) = 41%; ash = 7.6%; crude fiber < 0.5%; fat = 3.1%; mannan oligosaccharides (**MOS**) = 11%; glucan = 21%; thiamine (B1) = 29.3 mg/kg; riboflavin (B2) = 17.8 mg/kg; pantothenic acid (B5) = 22.8 mg/kg; pyridoxine (B6) = 14.6 mg/kg, cyanocobalamin (B12) = 56.3 µg/kg; N = 6.56%; Ca = 0.28%; P = 1.08%; Na = 0.37%; K = 2%; Mg = 0.17%; and zinc = 154 mg/kg). All cows were fed their respective diets once daily at 1400 h throughout the trial. The daily AY allocation was mixed with 300 g of ground corn and top-dressed onto the TMR immediately after feeding. Cows in LS0 and HS0 received a top dress consisting of 300 g of ground corn only. The total consumption of the top dress was verified daily. 

### 2.3. Data Collection and Sampling Procedures

Samples of TMR were obtained weekly and analyzed for DM [24] by drying for 24 h in a forced-air oven at 110 °C. The diet composition was adjusted weekly for changes in the DM content of the ingredients. The TMR offered and refused from each cow was recorded to determine intake based on weekly DM analyses. The total mixed ration samples were gathered before the addition of the top dress weekly through the experiment and stored at −20 °C until analyzed. Composite samples for each period (n = 5; 3 TMR samples per period) were analyzed for contents of DM, CP, acid detergent fiber (**ADF**), neutral detergent fiber (**NDF**), lignin, starch, fat, ash, Ca, P, Mg, K, Na, Fe, Zn, Cu, Mn, Mo, and S using wet chemistry methods (Dairy One, Ithaca, NY). Values for net-energy lactation (**NE_L_**) were provided by the lab and calculated based on NRC [25]. The physical characteristic of the TMR was recorded weekly utilizing the Penn State particle separator [26]. During the measurement phase, the TMR samples were collected to determine AD on d 18, 19, and 20, along with refusal collections of each period for each individual cow, and stored at −20 °C until analyzed for DM, NDF, CP, starch, ash, total N, and undegradable NDF at 120 h (**uNDF120**) (Dairy One, Ithaca, NY, USA; as described by Farmer et al. [27]). The N content of the AY (top dress) was not accounted for in the cow’s total N intake. 

Cows were milked three times daily at 0400, 1200, and 1930 h. Milk weights were recorded at every milking and samples were obtained at each milking on d 15 and 20 of each period. A preservative (800 Broad Spectrum Microtabs II; D&F Control Systems, Inc., San Ramon, CA, USA) was added to the milk samples collected on d 15 and 20. The preserved samples were stored in a refrigerator at 4 °C until they were composited in proportion to milk yield and sent to a commercial laboratory (Dairy One, Ithaca, NY, USA) to be analyzed for contents of fat, true protein, milk urea nitrogen (**MUN**), lactose, total solids, casein, and somatic cell count (**SCC**) using mid-infrared procedures [28]. 

Rumen fluid was collected via rumen cannula with a siphon on d 15 and 16 at 1400 h (time point 0), and at 4, 8, 12, 16, 20, and 24 h after feeding. Rumen fluid collection consisted of representative samples (ventral sac, cranial sac, and caudo-ventral blind sac) to assess the effects of the daily fluctuation of rumen pH (AP115, Fisher Scientific, Pittsburgh, PA, USA). At the same time, representative rumen fluid (500 mL) was collected. After collection, 20 mL of rumen fluid was strained through a four-layer cheesecloth combined with 20 mL of 2N HCl, kept at 4 °C for 24 h, and stored at −80 °C until analyzed for volatile fatty acid (**VFA**) proportions (Dairy One, Ithaca, NY, USA). On d 18 and 20, fecal pH was recorded by directly inserting the probe into the fecal content (AP115, Fisher Scientific, Pittsburgh, PA, USA). 

Urine samples (15 mL) were collected by manually stimulating urination at 0800 and 2030 h on d 20 of each period. The urine samples were immediately acidified after collection by pipetting 15 mL of urine into a specimen container containing 60 mL of 0.072N H_2_SO_4_ and stored at −20 °C. The urine samples were later thawed and composited by volume by cow and period and analyzed for creatinine, uric acid, urea N (Veterinary Medical Diagnostic Laboratory, MO), allantoin [29], and total N (Dairy One, Ithaca, NY, USA). The daily urinary volume and excretion of total N, urea N, uric acid, and allantoin were approximated from the urinary creatinine concentration using a creatinine excretion rate of 29 mg/kg of BW [30]. Fecal samples (120 mL, wet weight) were collected directly from the cow’s rectum on d 18, 19, and 20 at 0800 and 2030 h. On d 18 and 20, fecal pH was recorded directly into fecal matter (AP115, Fisher Scientific, Pittsburgh, PA, USA). The fecal samples were composited on an equal wet weight basis. Similar to Farmer et al. [27], uNDF120 was used as an internal marker and AD calculations were conducted by the ratio technique using the nutrient and indigestible NDF concentrations in the TMR and feces, adjusted for each cow based on the nutrient composition of the diet offered and refused [31].

Blood was sampled from the coccygeal vein or artery at 0600 h on d 15, 18, and 21 (n = 3) of each period from each cow (BD Vacutainer; BD and Co., Franklin Lakes, NJ, USA). Blood samples for plasma were collected into tubes containing heparin sulfate and placed on ice. Serum and plasma samples were obtained by centrifugation of the tubes at 2000× *g* for 15 min at 4 °C and stored at −80 °C for further analysis. Beta-hydroxybutyrate (**BHB**) was analyzed from whole blood immediately after sampling using a digital cow-side ketone monitor (Nova Max Plus, Nova Biomedical Corporation, Waltham, MA). Heparinized plasma samples (n = 225) were analyzed for bovine chemistry profiles [plasma urea nitrogen (**PUN**), total protein, albumin, globulin, Ca, P, Na, K, glucose, alkaline phosphate total, aspartate aminotransferase (**AST**), gamma-glutamyl transpeptidase (**GGT**), total bilirubin, creatine phosphokinase (**CPK**), total cholesterol, glutamate dehydrogenase (**GLDH**), and triglycerides] using the AU680 Beckman Coulter analyzer (https://vdl.vetmed.illinois.edu/clinical-pathology/; accessed on 1 July 2022) at the University of Illinois Veterinary Diagnostic Laboratory. Commercially available kits were used to analyze heparinized plasma samples for superoxidase dismutase (**SOD**; intra-assay CV = 6.8%, inter-assay CV = 13.5%), glutathione peroxidase (**GSH-Px**; intra-assay CV = 6.4%, inter-assay CV = 20.0%)**,** and lipopolysaccharide-binding protein (**LBP**; intra-assay CV = 5.8%, inter-assay CV = 12.9%). All kits were performed following the manufacturer’s instructions. Superoxidase dismutase activity was identified using a superoxidase dismutase assay kit; the dismutation of superoxide radicals generated by xanthine oxidase and hypoxanthine were measured (Cayman Chemical, Ann Arbor, MI, USA; https://www.caymanchem.com/search?q=706002; accessed on 1 July 2022). Glutathione peroxidase activity was analyzed using a glutathione assay kit using an indirect approach to measuring glutathione reductase for the quantification of glutathione reduction by GSH-Px (Cayman Chemical, Ann Arbor, MI, USA; https://www.caymanchem.com/search?q=703102; accessed on 1 July 2022). Plasma LBP was measured using a human lipopolysaccharide-binding protein multispecies reactive ELISA kit (Cell Sciences, Newburyport, MA, USA; https://www.cellsciences.com/human-lbp-elisa-kit-w-precoated-plates; accessed on 1 July 2022). Commercial kits were used to analyze EDTA plasma samples for non-esterified fatty acid (**NEFA**; intra-assay CV = 4.5%, inter-assay CV = 11.1%) and serum amyloid A (**SAA**; intra-assay CV = 4.8%, inter-assay CV = 14.9%). Non-esterified fatty acid analysis was performed using a (Wako Diagnostics U.S.A, Richmond, VA, USA; https://healthcaresolutions-us.fujifilm.com/in-vitro-diagnostics/clinical-diagnostic-reagents/non-esterified-fatty-acid-nefa; accessed on 1 July 2022) procedure followed by Johnson and Peters [32]. Serum amyloid A was analyzed using a phase range multispecies SAA ELISA kit (Tridelta Development, Ltd., Maynooth, Ireland; http://www.trideltaltd.com/Serum-Amyloid-A-Assay-Kit.html; accessed on 1 July 2022). Serum samples were analyzed using a D-lactate assay kit (Intra-assay CV = 4.5%, inter-assay CV = 15.2%; Cayman Chemical, Ann Arbor, MI, USA; https://www.caymanchem.com/search?q=700520; accessed on 1 July 2022), and a bovine insulin ELISA kit (intra-assay CV = 4.4%, inter-assay CV = 12.4%; Mercodia AB, Uppsala, Sweden; https://www.mercodia.se/mercodia-bovine-insulin-elisa; accessed on 1 July 2022). 

Health evaluations and visual assessments were performed for three consecutive days at the start of each period to monitor general appearance and fecal score. The rectal temperature was measured using a GLA M700 thermometer (GLA Agricultural Electronics, San Luis Obispo, CA, USA). The respiration rate was recorded by visually watching the cow breathe for 15 s, and the heart rate was measured via palpation of the femoral artery for 15 s. General appearance was scored using a similar method to Krause et al. [33], using a scoring system of 1 through 4: 4 = bright and alert; and 1 = down cow, without getting up. Fecal scores were allocated on a 1 through 4 scale according to Krause et al. [33]: 1 = runny, liquid consistency; and 4 = hard, dry appearance, original form not distorted on impact and settling. The body weight was measured (Ohaus digital scale, model CW-11, Newark, NJ, USA) and the body condition score (**BCS**) was assigned in quarter-unit increments for each cow weekly [34]. Three people independently assigned a BCS score and the average score was used for statistical analysis. 

### 2.4. Statistical Analyses

Data collected from d 15 to 21 of each period were analyzed using SAS (v. 9.4, SAS Institute Inc., Cary NC). For production variables and fecal pH, the MIXED procedure of SAS was used to model the fixed effects of treatment, square, and period:$$Y_{ijk}=\;\mu +T_{i}+S_{j}+P_{k}+T_{i}\times S{\;}_{j}+T_{i}\times P_{k}+S_{i}\times P_{k}+C{\left(S\right)}_{lj}+\varepsilon_{ijkl}$$
where $Y_{ijk}$ = the observations for dependent variables; μ = the overall mean; $T_{i}$ = the fixed effect of the *i*th treatment; $S_{j}$ = effect of the *j*th square; $P_{k}$ = effect of the *k*th period; $T_{i}\times S_{j}\;$= interaction between $T_{i}$ and $S_{j}$; $T_{i}\times P_{k}\;$= interaction between $T_{i}$ and $P_{k}$; $S_{j}\times P_{k}\;$= interaction between $S_{j}$ and $P_{k}$; $C{\left(S\right)}_{lj}$ = random effect of the *l*th cow nested within the *j*th square; and $\varepsilon_{ijkl}$ = the random residual error. The model was reduced if interactions were nonsignificant (*p* > 0.10). For variables measured over time (rumen pH, blood metabolites, and VFA), the above model was used with the addition of repeated measures to model the fixed effects of time and treatment by time interaction. Variables were subjected to five covariance structures: compound symmetry, unstructured, autoregressive order 1, autoregressive heterogeneous order 1, and Toeplitz. Compound symmetry was the covariance structure that yielded the lowest corrected Akaike information criterion and was used in the model [35]. For both models, the cow was the experimental unit and was considered as a random effect. The carryover effects from squares 1 and 2, 2 and 3, and 1 and 3 were calculated [36]. There was no carryover effect for any variable of interest (*p* > 0.12). Two single degree-of-freedom contrasts were used: LS0 compared with HS0, and HS0 compared with the average of HS15, HS30, and HS45. Linear and quadratic treatment effects for HS0, HS15, HS30, and HS45 were also evaluated. The values reported are the least squares means and associated standard errors of the mean. Covariates were included in the model when analyzing the dependent variables DMI, milk yield, BW, and BCS. The area under the curve was calculated based on the incremental area method [37] with pH = 5.6 as the base line. The degrees-of-freedom method was the Kenward–Rogers [35]. Residuals distribution was evaluated for normality and homoscedasticity. A log transformation was used for the variables SCC, CPK, SAA, and NEFA. A square root transformation was used for the variable Na in plasma. All transformations were performed for better homogeneity of the distribution of residuals. All means shown for these variables were back-transformed. A multivariable logistic mixed model (FREQ procedure) was used for the dichotomized variables (fecal score and general appearance). The chi-square was computed and is presented. General appearance scores ≥ 3 were categorized as healthy and ≤2 were categorized as abnormal. Fecal scores > 2 were categorized as healthy and ≤2 were categorized as abnormal [38,39]. Statistical significance was declared at *p* ≤ 0.05 and trends at 0.05 < *p* ≤ 0.10.

## 3. Results

### 3.1. Diet Composition

The ingredient composition of the diets are in Table 1. Analyzed nutrients from the experimental diets are in Table 2. 

The physical characteristics of the LS TMR, based on the Penn State particle separator [26], was (mean ± SD): 3.6 ± 1.4% on upper (19 mm pore size), 43.2 ± 3.9% on middle (8 mm pore size), 15.3 ± 1.2% on lower (4 mm pore size) sieves, and 37.8 ± 4.7% in the pan. The physical characteristics of the HS TMR was (mean ± SD) 2.2 ± 0.9% on upper (19 mm pore size), 32.4 ± 2.6% on middle (8 mm pore size), 14.3 ± 1.1% on lower (4 mm pore size) sieves, and 50.6 ± 3.4% in the pan. 

### 3.2. DMI, BW, BCS, and Lactation Performance

Performance data for the measurement phase are in Table 3. 

Cows in LS0 had lower DMI (*p* < 0.001), BW (*p* = 0.003), and DMI as a percentage of BW (*p* < 0.001) than cows in HS0. We observed a quadratic treatment effect for BW (*p* = 0.05) and a tendency for a quadratic treatment effect for DMI (*p* = 0.09). Cows in HS0 had greater milk yield (*p* < 0.001) and energy-corrected milk (**ECM**) yield (*p* = 0.03) and tended to have higher 3.5% fat-corrected milk (**FCM**) yield (*p* = 0.08) when compared to cows in LS0. We observed a tendency for a positive linear effect for the casein percentage (*p* = 0.07). Cows in HS0 had greater protein (*p* < 0.001) and casein percentage (*p* = 0.01), along with protein (*p* <0.001), casein (*p* = 0.002), and lactose yield (*p* = 0.004), compared to LS0. Cows in HS0 had lower milk fat percentage (*p* = 0.007) when compared to LS0. Cows in LS0 tended to have higher MUN (*p* = 0.09) compared to HS0. Feed conversion efficiency decreased for cows on the HS0 treatment (milk yield/DMI, FCM/DMI, and ECM/DMI; *p* = 0.008; *p* < 0.001, and *p* < 0.001, respectively) compared to LS0. We observed tendencies for quadratic treatment effects with the addition of AY (FCM/DMI, ECM/DMI; *p* = 0.09 and *p* = 0.07, respectively). We observed no difference among the treatments for fecal score *(p* = 0.4) and general appearance health parameters. From 75 fecal score observations, 71 were considered healthy. However, of the four cows in the abnormal category (loose manure), two cows were in HS0, one was in LS0, and one in HS30. The general appearance of the cows observed was considered healthy throughout the experiment. 

### 3.3. Rumen pH, Fecal pH and VFA

Rumen pH, fecal pH, and VFA response data are in Table 4. 

Cows in LS0 had greater rumen pH (*p* < 0.001) than cows in HS0. A tendency for a quadratic effect of treatment was present for rumen pH (*p* = 0.08). Rumen pH differed (*p <* 0.0001) over time, but we observed no treatment by time-point difference (*p* = 0.44; Figure 1).

Cows in HS0 had a nadir rumen pH *(p* < 0.001) more acidic than cows in LS0. Cows receiving AY had a less acidic nadir rumen pH than cows not receiving AY (HS0, *p* = 0.007). A positive linear treatment effect was observed for nadir pH (*p* = 0.002). Fecal pH was different between treatments LS0 and HS0; cows in the LS0 treatment had greater fecal pH (*p* = 0.04) when compared to HS0. Cows in HS0 had greater total VFA (**TVFA**; *p* < 0.001) when compared to LS0, as well as if compared to the average of HS15, HS30, and HS45 (*p* = 0.02). We observed quadratic (*p* = 0.03) treatment effects for TVFA. Cows in LS0 had lower (*p* < 0.007) propionate, butyrate, and valerate as a proportion of TVFA concentration when compared to cows in HS0. Cows in HS0 had higher (*p* < 0.05) acetate, butyrate, isobutyrate, valerate, and isovalerate as a proportion of TVFA concentration when compared to the average of cows in HS15, HS30, and HS45. There was a negative linear treatment effect (*p* < 0.04) for isobutyrate and isovalerate. In addition, there was a quadratic treatment effect (*p* < 0.03) for propionate, butyrate, and valerate, as well as a tendency (*p* = 0.07) for acetate. Propionate as a proportion of TVFA varied with time (*p* < 0.0001) and treatment × time (*p* < 0.0001) interaction (Figure 2). 

Treatments by day interactions were not present in other variables (*p* > 0.33).

### 3.4. Nitrogen Excretion and AD

The nitrogen excretion and AD data for the measurement phase are in Table 5.

Cows in HS0 had a lower milk protein N yield (*p* = 0.04) and a tendency (*p* = 0.08) for greater urea N, as a percentage of total urinary N and N intake yield, when compared to cows in LS0. Cows in HS0 tended to have greater allantoin (*p* = 0.08) and had greater (*p* = 0.03) total purine derivatives (**PD**) and microbial N production when compared to cows in LS0. A tendency for a quadratic treatment effect was present for urine allantoin (*p* = 0.08) and a quadratic treatment effect for uric acid (*p* = 0.01). We observed a tendency (*p* = 0.07) for a linear decrease for fecal N as a percentage of intake with AY supplementation. Cows in HS0 had greater (*p* ≤ 0.05) nutrient intakes for CP and starch, and tended (*p* = 0.06) to have greater nutrient intake for OM when compared to cows in LS0. Cows in LS0 tended to have greater AD of starch (*p* = 0.08) and NDF (*p* = 0.10) compared to HS0. A positive linear tendency was present for AD of CP (*p* = 0.09).

### 3.5. Serum and Plasma Chemistry Profile

The blood plasma and serum chemistry profiles are in Table 6. 

Cows in LS0 had greater (*p* < 0.02) concentrations for NEFA and total bilirubin, and lower (*p* < 0.03) concentrations for insulin, phosphorus, and GGT, in comparison to cows in HS0. Cows in LS0 tended (*p* = 0.07) to have greater concentrations of total protein and calcium than cows in HS0. Cows that received AY had higher (*p* < 0.05) total protein, globulin, calcium, and SAA, but a lower (*p* = 0.03) albumin/globulin ratio than cows that did not receive AY (HS0). Cows that received AY tended (*p* = 0.07) to have lower SOD concentrations than cows that did not receive AY (HS0). There was a negative linear treatment effect (*p* = 0.05) for SOD and a tendency (*p* = 0.08) for a negative linear treatment effect for BHB and triglycerides. Finally, there was a quadratic treatment effect (*p* < 0.03) for glucose, total protein, globulin, and calcium; and a tendency for albumin (*p* = 0.09), albumin/globulin ratio (*p* = 0.06), and sodium (*p* = 0.09) concentrations.

## 4. Discussion

This study postulated that the addition of an AY supplementation may improve the rumen environment, thus improving the efficiency of microbial N use, AD, and blood metabolites (i.e., SOD) when feeding HS diets, supporting lactation performance. This experiment was able to successfully achieve a high (31.2 ± 4.2% starch in DM) and low starch (22.8 ± 0.7% starch in DM) diet [3,4,5]. In the current study, corn grain was the main starch source used to achieve a HS diet. When formulating the LS diet, we decreased corn grain and increased the corn silage, reducing the starch content and increasing NDF. Our approach was aimed to prevent variable effects on rumen environment when using different feedstuffs to manipulate dietary starch. For instance, the inclusion of beet pulp in the LS diet and not in the HS diet for the purpose of starch concentration comparison allows for differences in the rumen microbiome [22]. Such an approach contributed to the nascence of carryover effects. To quantify our diets, we considered starch values in the low 20s as LS (<24) and in the high 20s as HS (>26) on a DM basis for this experiment and did not cause sub-acute ruminal acidosis (Table 4; [40]). 

In previous studies, no differences were reported for DMI for cows fed HS and LS concentrations [7,41]. Broderick et al. [3] fed mid-lactation cows diets containing 31% starch (HS) or 20% starch (LS; TMR DM basis) and the LS diet had reduced DMI. Similar results were observed in the current study. Yeast supplementation at greater concentrations (HS30 and HS45) tended to improve DMI. Cows in HS45 had the greatest DMI, which may explain why cows in the HS45 treatment had greater BW when compared to the other AY supplemented treatments. In the current study, cows in the HS0 treatment had greater BW when compared to LS0, and this difference may be attributable to the increase in DMI and a more energy-dense diet. Previous research reported that milk yield increases when feeding an HS diet in comparison to LS [3,17]. Observed increases in milk yield when feeding HS diets suggest that greater energy concentrations in the diet increase ruminal propionate, providing increased glucose for better utilization of milk yield when compared to LS diets. Additionally, cows in HS30 tended to increase milk yield with AY supplementation. This could be attributable to a combination of factors such as improved ruminal propionate and pH. Miettinen and Huhtanen [42] reported that an increase in ruminal butyrate supply at the expense of propionate adversely affected milk yield and the repartitioning of nutrients between milk components. A more in-depth observation of rumen characteristics will help determine the role of bacteria in the rumen on lactation performance. In contrast to the current study, Desnoyers et al. [16] conducted a meta-analysis concluding that milk yield increased with increased yeast dosage, dissimilar to the current study, at the HS45-dosage milk yield decreased when compared to HS30. 

An increase in milk yield for cows in HS0 may be responsible for the observed increase in ECM, and this could be initiated by the greater fat and protein yields. When comparing milk composition, Dias et al. [17] reported no differences between HS and LS diets for yields of FCM or ECM, nor with FCM/DMI or ECM/DMI feed conversion efficiencies. However, with the addition of yeast, Dias et al. [17] reported improvements in milk yield, ECM, and milk fat and protein yield when compared to the control diet. The present study reported improvements in milk components (ECM, fat, protein, and casein percentage, and lactose yield) at the HS30 dosage, but components declined as AY dosage increased. Furthermore, cows in the HS15 treatment tended to have improved efficiencies (FCM/DMI and ECM/DMI) in comparison to the other HS treatments. Feed efficiencies were greater in the LS0 diet compared to HS0. Ferraretto et al. [9] observed reduced FCM/DMI, ECM/DMI, and milk yield/DMI efficiency with reduced dietary starch (30.4 vs. 36.3% starch in DM), whereas Gencoglu et al. [43] reported no effect of starch (33.3 vs. 20.9% starch in DM) for the same efficiencies for cows with milk yields between 49.8 and 50.9 kg/d, suggesting that animal responses may vary with dietary ingredients and dietary starch content. 

Highly fermentable diets can negatively affect milk composition, particularly milk fat content. In the current study, an increase in milk fat concentration is observed for cows in LS0. Milk fat concentrations can be related to starch concentration, NDF, and ruminal digestion rates [44]. Additionally, increasing starch concentration can decrease the transfer of dietary polyunsaturated fatty acids to milk, increasing the pathways to adipose tissue instead [45]. In our experiment, the HS30 diet tended to have the greatest VFA acetate concentration when compared to the other HS diets supplemented with AY, contributing to milk fat yield. Cows in the LS0 treatment had increased forage NDF in their diets, which allowed for elevated ruminal pH compared to HS0. Furthermore, because cows in HS0 had greater rumen TVFA, propionate, and lower butyrate concentrations in the rumen than cows in LS0, this may have been associated with the suppressed milk-fat concentration of cows in HS0. A similar mechanism could have occurred for cows in HS0 related to low ruminal pH, as it favors increased propionate, lactic acid, and glucose production, which elevates insulin, thus reducing the free fatty acids released from adipose tissue, impacting milk fat [41,46]. Likewise, feeding HS (30% of starch DM) tended to decrease the milk-fat proportion compared to LS (20% of starch DM; [41]). Comparable results were seen in the current study, as cows receiving the HS0 diet had lower concentrations of milk fat. Nonetheless, the cows did not differ in the general appearance health parameters in the current study.

Both the current study and Dias et al. [17] reported greater milk protein concentrations and yields for HS diets compared to LS. Increased milk yields and milk protein concentrations for cows in HS0 may be attributed to the greater energy density of the diet. Previous research has reported a positive correlation between milk protein concentration and ruminal propionate yield [47]. Propionate production generates glucose through gluconeogenesis; once taken up by the mammary gland, glucose supports lactose synthesis [48]. As a result, we observed greater milk and lactose yield in the HS0 diet. The addition of AY tended to improve the casein percentage for the HS30 treatment, which is explained by the quadratic treatment response in the ruminal propionate when compared to HS0. Cows in LS0 tended to have increased MUN compared to cows in HS0. Gencoglu et al. [43] and Ferraretto et al. [9] reported similar results for MUN concentration. Increasing the dietary NDF concentration has been reported to increase ruminal ammonia, more so than increasing starch concentration [9]. This may explain the greater MUN response while feeding the LS0 treatment in the current study. The addition of rapid starch fermentation provided in HS0 may have allowed for increased nitrogen utilization by rumen bacteria, leading to an increase in microbial protein production. Additionally, with AY, the HS30 treatment has numerically improved MUN, which may suggest better N utilization. Slightly greater MUN values for the LS0 diet are consistent with reduced milk yield and true protein. 

Increasing the dietary starch concentration is known to decrease ruminal pH, as was observed in the current study for cows in HS0 compared to LS0 [49]. The addition of yeast attenuated shifts in the rumen environment, increasing rumen pH, comparable to AlZahal et al. [50]. A similar study conducted by Neubauer et al. [18] may explain a potential mechanism for the pH stabilization observed when feeding AY. Neubauer et al. [18] used the same AY product (15 g per cow per day) and reported an increase in cellulolytic bacteria, along with the reduction in starch-fermenting gram-positive bacteria by yeast when feeding HS concentrates. Nadir pH demonstrated a similar pattern, with HS0 having the lowest pH and pH increasing as the concentration of AY in the diet increased. There are fewer reports in the literature regarding fecal pH. According to Beasom et al. [51], fecal pH is a function of diet. However, rumen and fecal pH are usually not related unless starch bypasses the rumen and results in hindgut fermentation [40]. According to Gressley et al. [52], an observed variation in fecal consistency, like diarrhea, frothy feces, and mucin casts, would occur because of hindgut fermentation. In the present study, we observed no differences detected for fecal consistency per fecal score evaluation. Reducing NDF concentrations in the diet has been reported to increase the total rumen VFA concentration [53]. This was observed in the current study, as cows in HS0 had increased ruminal VFA concentration compared to cows in LS0.

Propionate concentration may influence DMI as it can be oxidized in the liver [46], decreasing intakes. If oxidation of propionate in the liver occurs, it would be a short-term effect and it is not likely for the current study as increased ruminal propionate improved DMI. Propionate and butyrate concentrations are most highly correlated to milk yield, which is positively correlated to DMI [54]. Cows in HS0 had greater propionate concentrations with greater milk yield and DMI. Gluconeogenic substrates, propionate and valerate, can both help synthesize lactose, the main contributor of milk yield [55]. These data may be supported in the current study, as an increase in milk yield was observed for cows in HS0 compared to LS0. Although a treatment by time interaction was noted for propionate, the variable had a large SEM and directional interaction, perhaps with limited biological relevance. 

All diet treatments were isonitrogenous (17.5 ± 0.56% CP, as a percentage of DM); however, cows in HS0 tended to have a greater N intake when compared to LS0. The greater N intake may have been due to cows in HS0 consuming greater DMI, also resulting in increased milk protein N when compared to the LS treatment. We did not observe increased N intake and milk protein N with increased AY and DMI. Aguerre et al. [56] reported 93% of the N intake by cows was in fecal N, milk N, or emitted as NH_3_-N. Moorby et al. [57] reported increased N intake, milk N, fecal N, and DMI with greater concentrate inclusion. However, Dias et al. [7] reported no differences in N intake, increased milk N for HS (29.0 ± 1.2% starch DM) versus LS (23.2 ± 1 7% starch DM) treatments, and increased milk N with yeast supplementation, but no difference for DMI across treatments. Thus, the increased milk N reported in the current study for HS diets may not just be a result of greater intakes but of changes in the microbial protein yield eliciting changes in the amino acid profile in the small intestine between the two diets [7].

Cows consuming HS had greater CP nutrient intakes, which can increase RDP into microbial protein, increasing N utilization efficiency [58]. Cows in HS0 tended to have greater total PD and microbial N production when compared to cows in LS0. The reason greater concentrate diets may increase microbial protein is because the additional energy substrate provided to ruminal microorganisms may improve the efficiency of fermentation in the rumen, thus enhancing the capture of ruminal ammonia into microbial protein [59]. Likewise, Valadares et al. [30] reported that total PD and microbial N increases as the concentrate-to-forage ratio increases (20 to 65 dietary concentrate, % of DM). 

Agle et al. [59] reported no treatment differences for urine allantoin, uric acid, total PD, or microbial protein when feeding a high concentrate (29.6% starch DM) versus a low concentrate (21.3% starch DM). However, the addition of AY tended to decrease urine allantoin and increased uric acid, but, contrary to the current study, Hristov et al. [60] reported a tendency for increased urine allantoin with yeast supplementation with no difference for uric acid. The LS0 treatment, assuming a greater prevalence in rumen fibrolytic bacteria compared to HS0, could aid in improving the efficiency of conversion of ruminal ammonia to microbial N production [60]; therefore, this may help explain the lower urea N as a percentage of total urinary N tendency, as urinary N is the main source of ammonia excretion from cattle. Fecal N efficiency (total N, as a percentage of N intake) tended to be linearly improved with the most efficiency at HS45. 

Fecal N excretion decreases with the yeast cell wall (**YC**) compared to no yeast supplementation [22] in the current study; fecal N excretion numerically decreased by 10% with AY, the lowest N excretion at HS15. Yeast supplementation may modulate changes in the rumen, increasing fibrolytic bacteria that have a high inclination for ammonia as a N source [61]; if this is the case, increased utilization of ammonia in the rumen for synthesis of microbial protein may be expected, improving overall dietary N efficiency [60]. An increase in fecal N efficiency by AY supplementation can have a positive impact on the environment [2,62]. 

Nutrient intakes and AD for cows fed HS and LS diets in the current study are similar to Lascano et al. [22], who reported that feeding HS (27.9% starch DM) had greater organic matter (**OM**) and starch nutrient intakes when compared to feeding LS (16.7% starch DM). Similar to the present work, Lascano et al. [22] reported no linear or quadratic treatment differences for the YC dose on nutrient intakes. However, with increased DMI at HS30 and HS45, increased nutrient intakes for CP and starch were observed with AY. In addition, similar to the present study, the AD for OM was not different between the HS and LS treatments [20]. Many studies have reported improved OM digestibility when *S. cerevisiae* was supplemented because of increased fiber digestion [16,22]. 

A study with treatments varying in the forage-to-concentrate ratio (80:20, 65:35, 50:50, and 35:65% of DM) on AD of OM, N, and starch were all unaffected by treatment [57]. Mwenya et al. [63] reported that cows supplemented with YC had numerically greater CP digestibility (74.5 vs. 75.9% CP), and Wohlt et al. [64] reported a CP digestibility tendency (73.8 vs. 75.4% CP) with yeast supplementation concurrent with the current study. Additionally, previous studies [7,65] reported significant increases in CP digestibility with YC supplementation. Moorby et al. [57] observed the amount of dietary N digested in the rumen increased with greater feed intake. Cows in the HS45 treatment had the greatest DMI, along with the greatest apparent CP digestibility. 

A study conducted by Farmer et al. [27] observed that decreasing the forage in the diet reduced the NDF AD, consistent with the results from the current study [57,66,67]. Agle et al. [59], in agreement with the present study, reported a lower apparent NDF digestibility in higher-concentrate (29.6% starch in DM) diets when compared to lower-concentrate (21.3% starch in DM) diets. Increasing the amount of rapidly fermentable starch in the diet while decreasing forage fiber can increase VFA production beyond the buffering and absorptive ability, thus reducing ruminal pH, which can have negative implications on fiber digestion, as observed in the present study [68]. In addition, starch and NDF AD tended to decrease in the HS0 diet compared to LS0; previous reports observed a positive linear increase in starch whole-tract digestibility with an increased concentrate [57]. According to Firkins et al. [69], on average, for every one-percentage unit increase of dietary NDF concentration, starch digestibility decreases approximately 0.6 percentage units. Prior studies have reported no starch treatment differences for high (32.9% starch in DM) vs. low concentrate (24.1% starch in DM; [66]). The decreased rumen pH observed when feeding the HS0 treatments may explain the decrease in starch digestibility when compared to the LS0 treatment. Subsequently, the enzyme activity of rumen fluid could affect starch digestibility [4].

Supplementation of AY in HS diets increased blood glucose concentration, similar to Lascano et al. [22]. Blood glucose can be an insensitive parameter of energy status because of its homeostatic regulation [70]. Glucose in blood plasma is not used by the liver; thus, it is not a source of energy for ruminant animals [71]. Similar to Oba and Allen [4], cows in HS0 had greater insulin concentrations, which is likely reflective of the increased energy density of the diet compared to LS0. A plasma marker more related to energy status and fat mobilization is NEFA because this metabolite responds more rapidly to changes in energy metabolism [72]. Dias et al. [7] reported no treatment effect for HS versus LS for plasma glucose or NEFA. While it is believed that cows in LS0 may be utilizing more fat deposits, releasing more free fatty acids and increasing the NEFA concentration, when compared to the HS0 diet. However, even though cows in HS0 had increased BW compared to LS0, the biological meaning of the change is questionable. The increased free fatty acids may be responsible for greater milk fat concentration in cows in LS0. The present study did not analyze fatty acids in milk, but further research could be conducted to validate this theory. Nevertheless, all NEFA concentrations were within the normal range [73,74].

Cows in the HS0 treatment experienced greater plasma phosphorus (**P**) concentrations compared to LS0. Feeding HS diets for an extended period can initiate rumen acidosis, thus facilitating bone demineralization and causing greater concentrations of plasma P for cows in HS0. However, it is important to report that plasma P concentrations for cows in both the LS0 and HS0 treatments were within the normal biological range (3.6–6.9 mg/dL; [73]). The plasma metabolites measured for liver functionality (AST, GGT, and total bilirubin and alkaline phosphate) were all within normal ranges (17 to 137 U/L, 4 to 28 U/L, 0.1 to 0.4 mg/dL, 8 to 179 U/L, respectively [73]). Gamma-glutamyl transferase is a membrane-bound enzyme located in many organs, such as the liver, and it can increase during signs of disease that can lead to liver damage [75]. Cows in HS0 had greater plasma GGT concentration compared to LS0, which may suggest more GGT passing through the liver when compared to LS0. The same interpretation could be made for total plasma bilirubin and the tendency for plasma AST to be higher in cows in LS0 than cows in HS0. This result can be interpreted as cows in LS0 having increased protein deamination and NEFA esterification that impacted in some degree on liver function [76]. 

A consequence of feeding HS diets is a decrease in rumen pH and the production of bacterial byproducts that can prompt inflammation [7]. Measuring acute-phase protein concentrations in the blood, such as albumin and SAA, can aid in detecting signs of inflammation. Acute-phase proteins are known as proteins that respond to inflammation by changing the blood concentration by >25% [77]. Bossaert et al. [78] reported that cows with increased inflammation post-calving had decreased albumin concentrations. Similarly, Burke et al. [79] diagnosed cows with endometritis as having decreased plasma albumin. In the current study, the addition of AY in HS diets resulted in a decreased plasma albumin concentration with increased AY, having the greatest concentration at HS15, suggesting an innate immune response is occurring with AY. Cows that received AY had higher SAA concentrations than cows not receiving AY. Elevation of SAA can occur under conditions unrelated to inflammation, such as physical stress [80]. During this study, there was no reason to believe cows in any one treatment experienced any additional stress compared to another treatment for the SAA concentrations to increase, as we observed no differences in fecal scores or general appearance. Cannizzo et al. [81] reported that cows at medium risk for acidosis (pH 5.77 ± 0.35) had the greatest SAA concentration, while high-risk cows with a lower ruminal pH (pH 5.57 ± 0.31) had a lower SAA concentration. Taken together, these data indicate that increased SAA concentration does not correspond to lower rumen pH values.

The concentration of the oxidative stress marker SOD decreased with increased AY supplementation. Superoxide dismutase aids in the defense against reactive oxygen species and antioxidant status. An elevated plasma SOD concentration for cows may indicate a physiological upgrading of the enzyme to help stabilize/negate radical challenges, while habituating animals to oxidative stress in aid of improving the antioxidant status [82]. Increased SOD when feeding HS may be a result of decreased ruminal pH or increased metabolic drive for milk production. High-yielding dairy cows with an increased milk drive have been reported to experience an increased production of free radicals and reactive oxygen species [83]. Similar to the current study, Abaker et al. [84] reported that cows fed high-grain diets had increased SOD plasma concentration than cows fed low-grain diets. However, in the current study, since the SOD concentration was within the normal range, it is likely that minimal oxidative stress occurred. Nonetheless, AY tended to reduce the plasma SOD concentration in cows, most likely because of the aforementioned improved antioxidant status.

## 5. Conclusions

Results from the present study, when using the same ingredients to increase dietary starch, indicated that HS diets increased DMI and milk yield, while having an impact on milk composition and lowering production efficiencies. Overall, supplementation of AY improved the rumen environment (i.e., VFA profile and rumen pH) when feeding HS diets. Additionally, supplementing dairy cow diets with AY tended to increase DMI, milk casein proportion, and feed conversion. Blood metabolites (NEFA and insulin) indicated that energy metabolism improved for cows in HS0 compared to LS0, with no major implications on liver function and inflammation biomarkers among the diets. In the current study, plasma SOD decreased linearly with the supplementation of AY, indicating an improved antioxidant status. Additionally, feeding HS0 increased CP and starch nutrient intakes and total urine PD while tending to decrease the starch and NDF AD in comparison to LS0. Interestingly, AY supplementation tended to increase CP AD and reduce fecal N excretion as a percentage of the intake, which may help alleviate environmental concerns about N excretion by lactating dairy cows. We observed that the best AY dosage for improvements of lactation performance and rumen health is at the lower AY concentrations (HS15, and HS30), and decreased benefits were observed with HS45. Similarly, the best AY dosage for improvements of digestibility, N utilization, and ruminal degradability is at the lower AY concentrations (HS15, and HS30).

## Figures and Tables

**Figure 1 animals-12-02445-f001:**
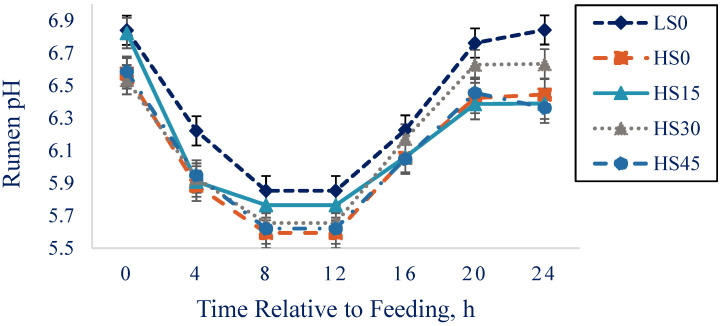
Least squares means (± SE) for rumen pH response to feeding (0 h) for cows in LS0, HS0, HS15, HS30, and HS45 treatments from 0 to 24 h relative to feeding (at 1400 h). Treatment: *p* < 0.0001; time: *p* < 0.0001; and no time × treatment interaction: *p* = 0.44.

**Figure 2 animals-12-02445-f002:**
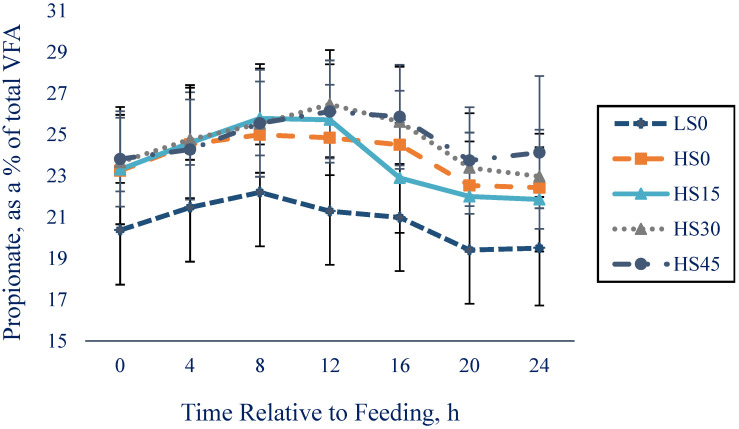
Least squares means (± SE) for propionate, as a percentage of total VFA response for cows in LS0, HS0, HS15, HS30, and HS45 treatments from 0 to 24 h relative to feeding (at 1400 h). Treatment: *p* < 0.0001; time: *p* < 0.0001; and treatment × time interaction: *p* < 0.0001.

**Table 1 animals-12-02445-t001:** Ingredient composition of the diets fed to lactating cows on a low-starch (LS) and high-starch (HS) diet during the experimental period.

	Diets	
Ingredient, % of DM	LS	HS
Corn silage ^1^	49.77	30.36
Alfalfa hay	16.02	17.38
Soybean meal	13.14	13.38
Dry ground corn grain	6.41	23.23
Canola meal	4.65	5.01
Corn gluten feed	2.69	2.91
Soy hulls	1.87	2.02
Dried molasses	1.38	1.49
Bypass fat ^2^	1.03	1.11
Dicalcium phosphate	0.40	0.44
Trace mineral ^3^	0.06	0.07
Rumen protected lysine ^4^	0.06	0.07
Rumen protected methionine ^5^	0.04	0.04
Potassium carbonate	0.13	0.13
Sodium bicarbonate	0.66	0.66
Calcium carbonate	0.65	0.65
Potassium chloride	0.17	0.17
Urea 46%	0.15	0.15
Salt, white	0.07	0.07
Magnesium oxide 54%	0.07	0.07
Vitamin and mineral mix ^6^	0.58	0.58

^1^ Corn silage was 29.3 ± 1.2% DM for all treatments. ^2^ Energy booster 100 (Milk Specialties Co., Eden Prairie, MN). ^3^ Availa Dairy (6.67% Zn, 3.34% Mn, 0.585% Cu, 0.167% Co; Zinpro Corp., Eden Prairie, MN). ^4^ Ajipro-L Generation 3 (Ajinimoto Heartland, Inc., Chicago, IL, USA). ^5^ Smartamine M (Adisseo, Alpharetta, GA, USA). ^6^ Vitamin and mineral mix was formulated to contain 13.50% Ca, 0.001% P, 3.92% salt, 10.90% Na, 6.68% Cl, 2.33% Mg, 8.27% K, 0.14% S, 1.77 mg/kg Co, 126.98 mg/kg Cu, 32.86 mg/kg I, 602.01 mg/kg Fe, 980.85 mg/kg Mn, 7.47 mg/kg Se, 3.15 mg/kg organic Se, 888.79 mg/kg Zn, 108.86 kIU/kg Vitamin A, 21.77 kIU/kg vitamin D_3_, 410.51 IU/kg vitamin E, 2.48 mg/kg choline, 18.21 mg/kg biotin, 0.16 mg/kg Niacin, 0.004 mg/kg thiamine.

**Table 2 animals-12-02445-t002:** Mean chemical composition and associated standard deviations for low-starch (LS) and high-starch (HS) diets fed to cows throughout the experimental period.

Item	LS		HS	
	Mean ^1^	SD	Mean ^1^	SD
DM, %	43.1	2.36	51.9	3.22
CP, % of DM	17.8	0.63	17.2	0.49
ADF, % of DM	21.4	1.37	18.6	1.06
NDF, % of DM	31.8	0.94	28.7	1.24
Lignin, % of DM	3.6	0.41	3.2	0.56
NFC, % of DM	37.5	1.89	42.5	1.52
Starch, % of DM	22.8	0.70	31.2	4.22
Crude fat, % of DM	3.9	0.29	3.8	0.26
Ash, % of DM	8.96	1.73	7.81	1.02
NE_L_, Mcal/kg of DM ^2^	1.63	0.04	1.69	0.04
Ca, % of DM	1.41	0.76	1.07	0.37
P, % of DM	0.44	0.01	0.45	0.01
Mg, % of DM	0.28	0.02	0.27	0.01
K, % of DM	1.57	0.09	1.47	0.04
Na, % of DM	0.33	0.05	0.31	0.01
S, % of DM	0.24	0.01	0.23	0.01
Fe, mg/kg	402	157.98	354	110.30
Zn, mg/kg	107	14.41	101	2.47
Cu, mg/kg	16	1.85	14	0.33
Mn, mg/kg	101	23.16	88	13.75
Mo, mg/kg	1.1	0.25	1.1	0.01

^1^ Mean diet composition of periods 1, 2, 3, 4, and 5 (n = 5).^2^ NRC [25].

**Table 3 animals-12-02445-t003:** Least squares means and associated SEM for BW, BCS, and production parameters of Holstein cows on low-starch diets without autolyzed yeast products (LS0), high-starch diets without autolyzed yeast products (HS0), and HS with 15 g (HS15), 30 g (HS30), and 45 g (HS45) of autolyzed yeast during the experimental period.

	Treatment ^1^						*p*-Value Contrasts ^2^			
Variable	LS0	HS0	HS15	HS30	HS45	SEM ^3^	LS0 vs. HS0	HS0 vs. HS15, 30, 45	Linear TRT	Quad TRT
DMI, kg/d	19.90	24.88	22.72	24.95	25.56	1.08	<0.001	0.61	0.25	0.09
BW, kg	665	689	671	681	685	8.2	0.003	0.11	0.94	0.05
DMI, % of BW	2.91	3.51	3.46	3.53	3.71	0.25	<0.001	0.69	0.20	0.31
BCS	3.41	3.48	3.47	3.41	3.44	0.06	0.29	0.46	0.39	0.68
Milk yield										
Milk yield, kg/d	30.50	34.51	32.42	33.93	33.75	1.38	<0.001	0.21	0.82	0.22
FCM, kg/d	31.85	34.36	32.30	34.74	33.12	3.13	0.08	0.40	0.78	0.82
ECM, kg/d	31.27	34.39	32.17	34.92	33.20	3.14	0.03	0.41	0.85	0.79
Milk composition										
Fat, %	3.89	3.56	3.78	3.56	3.60	0.17	0.007	0.35	0.85	0.30
Fat, kg/d	1.16	1.18	1.15	1.22	1.16	0.11	0.76	0.98	0.96	0.62
Protein, %	3.13	3.23	3.23	3.29	3.24	0.04	<0.001	0.16	0.11	0.17
Protein, kg/d	0.94	1.10	1.02	1.13	1.07	0.10	<0.001	0.42	0.88	0.62
Casein, %	2.61	2.66	2.66	2.72	2.68	0.03	0.01	0.13	0.07	0.18
Casein, kg/d	0.32	0.43	0.37	0.41	0.41	0.06	0.002	0.28	0.81	0.31
Casein, % of protein	82.06	82.42	82.11	82.59	82.60	0.43	0.29	0.96	0.33	0.50
Lactose, %	4.67	4.73	4.73	4.69	4.72	0.05	0.11	0.49	0.56	0.44
Lactose, kg/d	1.41	1.63	1.49	1.61	1.57	0.15	0.004	0.23	0.82	0.34
MUN, mg/dL	14.37	13.56	14.18	13.20	13.86	0.49	0.09	0.63	0.96	0.94
SCC × 1000/mL	205	205	175	216	221	113	0.99	0.95	0.18	0.24
Milk/DMI	1.55	1.38	1.40	1.39	1.34	0.11	0.002	0.98	0.43	0.34
FCM/DMI	1.62	1.32	1.42	1.39	1.31	0.11	<0.001	0.37	0.82	0.09
ECM/DMI	1.64	1.32	1.41	1.39	1.31	0.11	<0.001	0.34	0.86	0.07

^1^ Dietary treatments were low-starch diets (LS0, without autolyzed yeast (Saccharomyces cerevisiae) product), high-starch diets (HS0, without yeast), and 15 g (HS15), 30 g (HS30), and 45 g of yeast (HS45) in a top dress. The top-dress vehicle was 300 g of ground corn. The LS0 and HS0 treatments received 300 g of corn in a top dress. ^2^ The contrasts were LS0 compared with HS0; HSO compared with the average of HS15, HS30, and HS45; and the linear and quadratic effects of treatments (HS0, HS15, HS30, and HS45). ^3^ Greatest value within treatment standard error of the mean.

**Table 4 animals-12-02445-t004:** The least squares means and associated standard errors for rumen pH, fecal pH, and VFA response in a low-starch diet without an autolyzed yeast product (LS0), a high-starch diet without an autolyzed yeast product (HS0), HS with 15 g (HS15), 30 g (HS30), and 45 g (HS45) of autolyzed yeast.

	Treatment ^1^						*p*-Value Contrasts ^2^			
Variable	LS0	HS0	HS15	HS30	HS45	SEM ^3^	LS0 vs. HS0	HS0 vs. HS15, 30, 45	Linear TRT	Quad TRT
Rumen fluid										
Ph ^4^	6.38	6.10	6.15	6.18	6.12	0.05	<0.001	0.17	0.54	0.08
pH < 5.6, h ^5^	6.28	7.15	5.97	6.62	8.08	1.50	0.62	0.84	0.49	0.23
Nadir pH	5.74	5.53	5.55	5.57	5.57	0.03	<0.001	0.007	0.002	0.43
AUC, pH × h/d ^6^	0.07	0.26	0.12	0.15	0.11	0.12	0.23	0.22	0.25	0.60
Fecal pH	6.95	6.71	6.72	6.59	6.66	0.08	0.04	0.58	0.43	0.75
Total VFA, mmol/L	127.63	137.42	129.75	134.08	134.43	2.96	<0.001	0.02	0.57	0.03
Individual VFA, mol/100 mol of total VFA ^7^										
Acetate	82.93	85.30	80.44	82.79	82.08	1.77	0.16	0.006	0.15	0.07
Propionate ^8^	20.75	23.87	23.74	24.63	24.79	2.60	<0.001	0.78	0.03	0.02
Butyrate	13.81	14.92	13.71	14.24	14.93	0.40	0.006	0.04	0.66	<0.001
Isobutyrate	1.05	1.04	0.98	0.99	0.95	0.03	0.41	<0.001	<0.001	0.44
Valerate	1.78	2.13	1.93	2.02	2.09	0.12	<0.001	0.04	0.97	0.005
Isovalerate	0.78	0.80	0.75	0.77	0.74	0.02	0.37	<0.001	0.004	0.21

^1^ Dietary treatments were low-starch diets (LS0, without autolyzed yeast (*Saccharomyces cerevisiae)* product), high-starch diets (HS0, without yeast), 15 g (HS15), 30 g (HS30), and 45 g of yeast (HS45) in a top dress. The top-dress vehicle was 300 g of ground corn. The LS0 and HS0 treatments received 300 g of corn in a top dress. ^2^ The contrasts were LS0 compared with HS0; HSO compared with the average of HS15, HS30, and HS45; and the linear and quadratic effects of treatments (HS0, HS15, HS30, and HS45). ^3^ Greatest value within treatment standard error of the mean. ^4^ Time point: *p* < 0.0001; treatment × time point: *p* = 0.44 (Figure 1). ^5^ Time points (TP) 0, 4, 8, 12, 16, 20, and 24 h relative to feeding at 1400 h. ^6^ Negative incremental area under the curve. Baseline rumen pH = 5.6. ^7^ Time: *p* < 0.0001; treatment × time: *p* > 0.33 for acetate, butyrate, isobutyrate, valerate, and isovalerate. ^8^ Time: *p* < 0.0001; treatment × time: *p* < 0.0001(Figure 2). All other variables, TRT × DAY interaction, were not present (*p* > 0.25).

**Table 5 animals-12-02445-t005:** Least squares means and associated SEM of nitrogen excretion and apparent digestibility in a low-starch diet without autolyzed yeast products (LS0), high-starch diet without autolyzed yeast products (HS0), HS with 15 g (HS15), 30 g (HS30), and 45 g (HS45) of autolyzed yeast during the experimental period.

	Treatment ^1^						*p*-Value Contrasts ^2^			
Variable	LS0	HS0	HS15	HS30	HS45	SEM ^3^	LS0 vs. HS0	HS0 vs. HS15, 30, 45	Linear TRT	Quad TRT
N intake, g/d	617	717	686	729	738	42	0.08	0.98	0.56	0.64
Milk protein N ^4^, g/d	140.29	165.99	149.65	176.40	170.45	16.61	0.04	0.95	0.28	0.53
Milk protein N, % of N intake	22.82	22.60	22.15	24.27	22.75	1.96	0.88	0.70	0.58	0.61
Urinary excretion										
Urine volume ^5^, L/d	39.87	37.03	38.01	36.65	40.94	3.01	0.43	0.59	0.34	0.49
Total N, g/d	244.30	263.31	250.26	245.08	270.49	15.74	0.30	0.57	0.77	0.13
Total N, % of N intake	41.33	35.10	42.30	34.54	38.45	3.51	0.14	0.31	0.86	0.56
Urea N, g/d	199.60	216.54	212.04	213.30	227.22	10.61	0.14	0.91	0.34	0.24
Urea N, % of total urinary N	76.92	82.41	81.87	80.15	83.62	2.31	0.08	0.83	0.84	0.36
Allantoin, mmol/d	172.87	194.81	170.21	181.85	186.47	12.14	0.07	0.11	0.72	0.08
Uric acid, mmol/d	66.58	74.32	60.46	73.61	78.91	5.57	0.16	0.43	0.11	0.01
Total PD, mmol/d	219.22	256.28	226.62	240.16	245.12	20.33	0.03	0.15	0.69	0.14
Microbial N production ^6^, g/d	137.97	161.30	142.63	151.15	154.27	12.79	0.03	0.16	0.81	0.22
PUN, mg/dL	14.90	14.22	15.54	13.93	15.27	0.57	0.38	0.26	0.53	0.99
Fecal N excretion										
N, g/d	221.95	250.09	239.35	260.48	242.38	17.51	0.25	0.73	0.93	0.99
N, % of intake	38.02	37.06	36.42	35.30	33.11	1.74	0.68	0.24	0.07	0.63
Nutrient intakes, kg/d										
OM	18.61	22.51	20.55	22.07	22.87	1.28	0.06	0.58	0.54	0.21
CP	3.66	4.50	4.08	4.59	4.61	0.30	0.05	0.76	0.48	0.38
Starch	4.71	6.53	6.32	6.95	6.94	0.42	0.002	0.65	0.30	0.81
NDF	6.15	6.54	6.03	6.11	6.45	0.36	0.88	0.97	0.99	0.96
Apparent digestibility, %										
OM	65.41	66.23	67.66	65.95	68.19	1.22	0.63	0.44	0.43	0.75
CP	61.13	62.04	64.05	63.33	66.07	1.67	0.73	0.15	0.09	0.91
Starch	95.43	94.13	94.77	93.72	94.72	0.54	0.08	0.64	0.75	0.72
NDF	52.03	47.87	51.86	45.36	50.81	1.75	0.10	0.46	0.76	0.66

^1^ Dietary treatments were low-starch diets (LS0, without autolyzed yeast (Saccharomyces cerevisiae) product), high-starch diets (HS0, without yeast), 15 g (HS15), 30 g (HS30), and 45 g of yeast (HS45) in a top dress. The top-dress vehicle was 300 g of ground corn. LS0 and HS0 treatments received 300 g of corn in a top dress. ^2^ The contrasts were LS0 compared with HS0; HSO compared with the average of HS15, HS30, and HS45; and the linear and quadratic effects of treatments (HS0, HS15, HS30, and HS45). ^3^ Greatest value within the treatment standard error of the mean. ^4^ Milk true protein N (milk true protein ÷ 6.38). ^5^ Estimated from creatinine concentrations in spot urine samples assuming a creatinine excretion of 29 mg/kg BW [30]. ^6^ Based on excretion of urinary purine derivatives [30].

**Table 6 animals-12-02445-t006:** Least squares means and associated SEM for blood metabolites of Holstein cows in low-starch diets without autolyzed yeast products (LS0), high-starch diets without autolyzed yeast products (HS0), HS with 15 g (HS15), 30 g (HS30), and 45 g (HS45) of autolyzed yeast during the experimental period. Samples were collected on d 15, 18, and 21 during the last week of each period.

	Treatment ^1^						*p*-Value Contrasts ^2^			
Variable	LS0	HS0	HS15	HS30	HS45	SEM ^3^	LS0 vs. HS0	HS0 vs. HS15, 30, 45	Linear TRT	Quad TRT
Blood ^4^										
Metabolism										
Glucose, mg/dL	70.15	70.14	71.90	71.53	70.52	1.04	0.99	0.11	0.79	0.03
GLDH ^5^, U/L	32.20	31.47	33.05	32.27	32.93	3.96	0.68	0.25	0.39	0.43
Cholesterol total, mg/dL	169	174	171	173	178	27	0.12	0.92	0.19	0.17
BHB, mmol/L	0.58	0.52	0.52	0.48	0.47	0.03	0.12	0.17	0.08	0.77
Triglycerides, mg/dL	8.18	7.81	8.18	8.43	7.83	0.59	0.36	0.28	0.80	0.08
NEFA ^6^, µEq/L	102.2	82.6	111.8	85.1	81.3	9.3	0.02	0.57	0.71	0.19
Insulin, µg/L	0.74	0.93	0.86	0.95	0.92	0.09	0.03	0.77	0.84	0.76
CPK ^7^, U/L	150	181	143	182	159	25.4	0.36	0.46	0.78	0.75
D-Lactate, m*M*	0.57	0.57	0.63	0.57	0.60	0.03	0.97	0.39	0.98	0.55
Total protein, g/dL	7.57	7.45	7.65	7.64	7.56	0.11	0.07	0.001	0.11	0.002
Albumin, g/dL	3.38	3.34	3.39	3.33	3.30	0.04	0.29	0.91	0.07	0.09
Globulin, g/dL	4.15	4.08	4.24	4.31	4.22	0.13	0.29	<0.001	0.01	0.005
Albumin/Globulin ratio	0.82	0.83	0.81	0.79	0.81	0.03	0.37	0.03	0.07	0.06
Minerals										
Calcium, mg/dL	9.31	9.12	9.29	9.34	9.20	0.09	0.07	0.05	0.37	0.03
Phosphorus, mg/dL	5.38	5.86	5.85	5.74	5.80	0.15	0.004	0.64	0.60	0.74
Sodium, mmol/L	136.52	135.78	136.71	136.60	134.84	0.68	0.44	0.58	0.07	0.09
Potassium, mmol/L	4.39	4.38	4.44	4.41	4.40	0.06	0.94	0.53	0.89	0.50
Na:K Ratio	31.13	31.11	30.96	31.07	30.80	0.42	0.98	0.66	0.59	0.87
Liver function										
AST ^8^, U/L	72.62	69.49	68.62	69.33	66.88	4.31	0.06	0.36	0.16	0.49
GGT ^9^, U/L	24.81	26.11	26.02	25.20	25.64	1.15	0.02	0.21	0.14	0.43
Total bilirubin, mg/dL	0.14	0.12	0.12	0.11	0.12	0.009	0.01	0.71	0.67	0.38
Alkaline phosphate total, U/L	46.03	46.32	43.69	46.69	45.22	2.39	0.78	0.18	0.93	0.42
Inflammation										
SAA ^10^, µg/mL	142	108	161	140	150	25	0.16	0.03	0.16	0.19
LBP ^11^, µg/mL	20.1	21.9	20.6	21.3	20.5	2.3	0.43	0.54	0.59	0.90
SOD ^12^, U/mL	3.67	4.08	3.77	3.60	3.50	0.32	0.19	0.07	0.05	0.64
GSH-Px ^13^, nmol/min/mL	86.9	84.1	87.7	81.5	88.8	4.2	0.55	0.61	0.60	0.59

^1^ Dietary treatments were low-starch diets (LS0, without autolyzed yeast (*Saccharomyces cerevisiae*) products), high-starch diets (HS0, without yeast), 15 g (HS15), 30 g (HS30), and 45 g of yeast (HS45) in a top dress. The top-dress vehicle was 300 g of ground corn. LS0 and HS0 treatments received 300 g of corn in a top dress. ^2^ Contrasts were LS0 compared with HS0; HSO compared with the average of HS15, HS30, and HS45; and the linear and quadratic effects of treatments (HS0, HS15, HS30, and HS45). ^3^ Greatest value within treatment standard error of the mean. ^4^ Time (day) differed (*p* < 0.05) for BHB, D-Lactate, potassium, Na:K, AST, and GSH-Px. Treatment × time interaction was not present for any variable (*p* > 0.17). ^5^ Glutamate dehydrognase. ^6^ Non-esterified fatty acid. ^7^ Creatine phosphokinase. ^8^ Aspartate aminotransferase. ^9^ Gamma-glutamyl transpeptidase. ^10^ Serum amyloid A. ^11^ Lipopolysacchride binding protein. ^12^ Superoxide dismutase. One unit (U) is defined as the amount of enzyme needed to exhibit 50% dismutation of the superoxide radical. ^13^ Glutathione peroxidase activity. One unit (nmol/min) is defined as the amount of enzyme that will cause the oxidation of 1.0 nmol of NADPH to NADP^+^ per minute at 25 °C.

## Data Availability

The datasets produced and/or analyzed during the current study are not publicly available because of confidentiality but are available from the corresponding authors upon reasonable request.

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
