# Peer review of "Effects of Autolyzed Yeast Supplementation in a High-Starch Diet on Rumen Health, Apparent Digestibility, and Production Variables of Lactating Holstein Cows"

_animals, 2022, doi:10.3390/ani12182445_

Round 1

Reviewer 1 Report

 I have some minor edits.

151-152 what about uric acid

166 plasma 

Author Response

151-152 what about uric acid

AU: good catch. Uric acid was added.

166 plasma

AU: thanks. This was corrected in the text.

Reviewer 2 Report

The manuscript : Effects of autolyzed yeast supplementation in a high starch diet on rumen health, apparent digestibility, and production variables of lactating Holstein cows;  present important data in the field of dairy cows nutrition and health. The experiment was well designed and conducted. 

I would suggest to add the chemical analysis of autolyzed yeast supplement , and add information in M&M section if the autolyzed yeast supplement N content were added to the total N intake of cows in the autolyzed yeast treatments. 

 Specific comment:

L 29: change - than ; to - compared to cows in. 

L 31: change - than ; to - compared to

L 108: delete extra space after full stop. 

L 244: delete extra space after full stop. 

Author Response

The manuscript : Effects of autolyzed yeast supplementation in a high starch diet on rumen health, apparent digestibility, and production variables of lactating Holstein cows;  present important data in the field of dairy cows nutrition and health. The experiment was well designed and conducted.

AU: Thanks.

I would suggest to add the chemical analysis of autolyzed yeast supplement , and add information in M&M section if the autolyzed yeast supplement N content were added to the total N intake of cows in the autolyzed yeast treatments.

AU: Per reviewer’s suggestions, we have added more information regarding the autolyzed yeast in the Materials and Methods section. The N content of the AY was not accounted to the total N intake of cows. This was clarified in the text.

 Specific comment:

L 29: change - than ; to - compared to cows in.

AU: This was corrected in the text.

L 31: change - than ; to - compared to

AU: This was corrected in the text.

L 108: delete extra space after full stop.

AU: Good catch. This was corrected in the text.

L 244: delete extra space after full stop.

AU: Good catch. This was corrected in the text.

Reviewer 3 Report

I would like to thank the Editorial Staff of Animals for entrusting me with the review of the Article Manuscript ID: Animals - 1824391 entitled: Title: Effects of autolyzed yeast supplementation in a high starch diet on rumen health,  apparent digestibility, and production variables of lactating Holstein cows.

Authors: Sara E. Knolinger, Milaine Poczynek, Bryan Miller, Isabel Mueller, Rodrigo De  lmeida, Michael R. Murphy, Felipe C. Cardoso

Submitted to section: Animal Physiology

Dear authors,

The manuscript is interesting, well written and discussed

There are several major issues that need to be addressed.

I would like to point out that the article is a continuation of other research in this field from; 2018, 2019.

e.g.

Knollinger, Sara Elizabeth

 “Effects of autolyzed yeast (saccharomyces cerevisiae) on feed degradability and digestibility; performance, rumen environment, and physiological biomarkers of Holstein cows fed a high starch diet”

The use of yeast in the nutrition of ruminants (dairy cattle, sheep, goats, broilers) is becoming an important issue due to the improvement of aging and the use of nutrients in highly productive animals. In ruminants, autolysed yeast is supposed e.g. to improve the number of cellulolytic bacteria in the rumen, or increase the energy level in the feed, and thus may affect the performance of animals.

The article contains a lot of valuable information from the obtained research, but it is prepared on a small number of animals (we do not know, among others, where do the animals come from?, what is their genetics?,…) – more information

My comments:

The introduction is written succinctly and synthetically; should be expanded to include the type of yeast used in research and trends therein ...

 In order to clearly define the type of AY commercially available i.e. name and trade number where it is mainly used in animal nutrition ...

Correctly planned experiment (in my opinion, although the research should be repeated), physicochemical analyzes adequate and correct,

 P:77- Should the knowledge about the dosage of yeast for ruminants be supplemented, as well as their production and physiological effects?

P:98-109- provide more information about selecting animals for experiment, why such BW discrepancies; why only 15 animals?

Were duplicates made? If not, should it be repeated in the future?

Has there been full approval for test procedures of this type on animals procedure number?

Why is such a share of yeast (AY) in HS15 doses; HS30; HS45?

P:161 – Explain it more?

P: 252 if there were significant differences in feed composition. Why such a high volatility - incl. Fe, Mn, Zn, Mo ?? in groups and between groups - no explanation in the text.

P: 254-258 check calculations (%)?

P: 278 add what were the abnormal symptoms in the cows

P: 283 - Table 4 results are not discussed or Table 4 is omitted in the text?

P: 373-374 explain why such ranks of DM - DM <24 is known to cause acidosis in cows

P: 389 better to interpret ...

P: 392-393 what could be the cause?

P: 408 the productivity level of the cows should be included in the studies quoted

P: 421-424 symptoms that may suggest the direction of chronic acidosis. Report the symptoms of the cows observed.

P: 449 list what doses of AY were used in nutrition

Conclusions

Conclusions are given generally to be interpreted for practice.

Consider which HS values ​​are most suitable for production, physiological and environmental purposes.

The literature follows the direction of the research and is correctly cited.

Summing up, I can say that due to the interesting results obtained, sometimes controversial the work should be accepted for publication after taking into account the reviewer's suggestions.

Kind regards,

Author Response

I would like to thank the Editorial Staff of Animals for entrusting me with the review of the Article Manuscript ID: Animals - 1824391 entitled: Title: Effects of autolyzed yeast supplementation in a high starch diet on rumen health,  apparent digestibility, and production variables of lactating Holstein cows.

Authors: Sara E. Knolinger, Milaine Poczynek, Bryan Miller, Isabel Mueller, Rodrigo De  lmeida, Michael R. Murphy, Felipe C. Cardoso

Submitted to section: Animal Physiology

Dear authors,

The manuscript is interesting, well written and discussed

There are several major issues that need to be addressed.

I would like to point out that the article is a continuation of other research in this field from; 2018, 2019.

e.g.

Knollinger, Sara Elizabeth

 “Effects of autolyzed yeast (saccharomyces cerevisiae) on feed degradability and digestibility; performance, rumen environment, and physiological biomarkers of Holstein cows fed a high starch diet”

The use of yeast in the nutrition of ruminants (dairy cattle, sheep, goats, broilers) is becoming an important issue due to the improvement of aging and the use of nutrients in highly productive animals. In ruminants, autolysed yeast is supposed e.g. to improve the number of cellulolytic bacteria in the rumen, or increase the energy level in the feed, and thus may affect the performance of animals.

AU: The submitted manuscript represents original research. Part of the results may have been presented at the American Dairy Science Association Annual meetings in the referred years. The present manuscript is not under consideration by any other journal.

The article contains a lot of valuable information from the obtained research, but it is prepared on a small number of animals (we do not know, among others, where do the animals come from?, what is their genetics?,…) – more information.

AU: Thanks. We understand the reviewer’s concern. Since this is a Latin Square design, this would be the equivalent of having 15 cows per treatment in a continuous experimental design (i.e., randomized complete block design). Our group has published similar experiments with this design, with a similar or lower number of animals (Sulzberger et al., 2016 – JDS 99:8028-8040; Sulzberger et al., 2017 – JDS 100:1856-1869; Pate et al., 2018 – JDS 101:11421-11434). Since we report significant treatment differences, we believe the number of experimental units was not an issue. We inform the reader upfront (in the title) that the work was performed on Holstein cows. We added information regarding the healthy state of the cows. Additionally, considering “cow” as a random effect in our model allows for such genetic variation to be accounted for.

My comments:

The introduction is written succinctly and synthetically; should be expanded to include the type of yeast used in research and trends therein ...

AU: We believe the section within lines 60 – 79 reports the effects of different types of AY and their effect. Nonetheless, we have expanded that section per the reviewer’s request.   

 In order to clearly define the type of AY commercially available i.e. name and trade number where it is mainly used in animal nutrition ...

AU: We have expanded information regarding the commercially available AY used in this experiment in the text.     

Correctly planned experiment (in my opinion, although the research should be repeated), physicochemical analyzes adequate and correct,

AU: Thanks. We believe the experiment has a solid experimental design that was already performed by our group (Sulzberger et al., 2016 – JDS 99:8028-8040; Sulzberger et al., 2017 – JDS 100:1856-1869; Pate et al., 2018 – JDS 101:11421-11434).      

 P:77- Should the knowledge about the dosage of yeast for ruminants be supplemented, as well as their production and physiological effects?

AU: The connection between our results and the published literature, including dosage, is done in the Discussion section. We would like to keep the Introduction section to introduce the topic and not exhaust the information like a literature review.

P:98-109- provide more information about selecting animals for experiment, why such BW discrepancies; why only 15 animals?

AU: We have included the information that all cows enrolled in the trial were healthy. The differences in BW pertain to a normal variation of dairy cows going through lactation (reported days in milk; DIM). The important factor to control is that all cows were past the peak of production to avoid confounding factors as they go through the Latin Square design. We have reported similar results previously (Sulzberger et al., 2016 – JDS 99:8028-8040; Sulzberger et al., 2017 – JDS 100:1856-1869; Pate et al., 2018 – JDS 101:11421-11434).  As mentioned before, since this is a Latin Square design, this would be the equivalent of having 15 cows per treatment in a continuous experimental design (i.e., randomized complete block design). Since we report significant treatment differences, we believe the number of experimental units was not an issue.

Were duplicates made? If not, should it be repeated in the future?

AU: In the Latin Square Design that we used in the present experiment, each cow received all the treatments and consequently allowed us to eliminate the “cow effect” as a source of variation. Therefore, the design has replication and allows for statistical inferences. We believe that repeating this experiment is not necessary but continuing to explore the effects of feeding AY to dairy cattle (i.e. effects on the rumen microbial population) should be beneficial for the scientific community.

Has there been full approval for test procedures of this type on animals procedure number?

AU: All experimental procedures were approved by the University of Illinois (Urbana-Champaign) Institutional Animal Care and Use Committee (#17172). This information was added to the text.

Why is such a share of yeast (AY) in HS15 doses; HS30; HS45?

AU: Our experiment aimed to explore the treatment space (i.e., dosage) from 0 to 45 g and explore the linear and quadratic effects of AY. This approach resulted on the conclusion that the dosage of 30g was the most beneficial dosage for most of the outcome variables (i.e. rumen fluid pH).    

P:161 – Explain it more?

AU: We believe the explanation for the blood collection procedure is complete and reported in a similar way elsewhere (Sulzberger et al., 2016 – JDS 99:8028-8040; Sulzberger et al., 2017 – JDS 100:1856-1869; Pate et al., 2018 – JDS 101:11421-11434). Nonetheless, we have added additional information per the reviewer’s request.

P: 252 if there were significant differences in feed composition. Why such a high volatility - incl. Fe, Mn, Zn, Mo ?? in groups and between groups - no explanation in the text.

AU:  The variation in the chemical analysis of the total-mixed ration (TMR) fed to cows is typical to be found when feeding dairy cows. The forages that are part of the TMR vary with time since, for instance, they may be coming from a different section of the field. Similar variation is reported by our group elsewhere (Fehlberg et al., 2020 – JDS 103:11386–11400; Sulzberger et al., 2016 – JDS 99:8028-8040; Sulzberger et al., 2017 – JDS 100:1856-1869; Pate et al., 2018 – JDS 101:11421-11434). Therefore the variation (SD) is not discussed in the manuscript.      

P: 254-258 check calculations (%)?

AU: The calculations were checked and are correct.

P: 278 add what were the abnormal symptoms in the cows

AU: The information was added to the text.

P: 283 - Table 4 results are not discussed or Table 4 is omitted in the text?

AU: Table 4 was discussed from lines 445 – 473 from the original version of the submitted manuscript. We have updated in the text.

P: 373-374 explain why such ranks of DM - DM <24 is known to cause acidosis in cows

AU: We are referring to the starch concentration on a DM basis. Sub-acute acidosis is characterized as rumen fluid pH of 5.2 and 5.6 for at least 3 h per day. We report our data for rumen fluid pH in Table 4. We have added a reference for Table 4 and sub-acute acidosis in the text. Additionally, the following paragraph further discusses the topic.

P: 389 better to interpret ...

AU: We have expanded the rationale in the text.     

P: 392-393 what could be the cause?

AU: We mention in the text that a more in-depth observation of rumen characteristics will help determine the role of bacteria in the rumen on lactation performance. Therefore we believe we would be speculating too much in trying to determine the causes for milk to AY response in both experiments.

P: 408 the productivity level of the cows should be included in the studies quoted

AU: The productivity level was added to the text.   

P: 421-424 symptoms that may suggest the direction of chronic acidosis. Report the symptoms of the cows observed.

AU: Cows in the present study did not differ in general appearance health parameters. This information was added to the text.

P: 449 list what doses of AY were used in nutrition

AU: Information was included in the text.  

Conclusions

Conclusions are given generally to be interpreted for practice.

Consider which HS values are most suitable for production, physiological and environmental purposes.

AU: We have included in the text the AY concentrations that are more suitable for production, physiological, and environmental purposes.  

The literature follows the direction of the research and is correctly cited.

AU: Thanks.

Summing up, I can say that due to the interesting results obtained, sometimes controversial the work should be accepted for publication after taking into account the reviewer's suggestions.

AU: Thanks. We believe we have addressed all the reviewer’s concerns.